# Efficient and stable inverted perovskite solar cells enabled by homogenized PCBM with enhanced electron transport

Cheng Gong [1,3], Haiyun Li[1,3], Zhiyuan Xu[1,3], Yuheng Li[2,3], Huaxin Wang[1], Qixin Zhuang[1], Awen Wang[2], Zhijun Li[1], Zhihao Guo[1], Cong Zhang[1], Baiqian Wang[1], Xiong Li [2] ✉ & Zhigang Zang [1] ✉

Fullerene derivatives are extensively employed in inverted perovskite solar cells due to their excellent electron extraction capabilities. However, [6,6]-phenyl-$C_{61}$-butyric acid methyl ester (PCBM) agglomerates easily in solution and exhibits a relatively low ionization barrier, increasing charge recombination losses and charge accumulation in the interface. Here, tetramethylthiuram disulfide (TMDS) is introduced into the PCBM solution to induce the formation of reducing sulfur radicals through UV light irradiation, allowing for n doping of the PCBM material. The resulting modified PCBM layer exhibits enhanced conductivity and electron mobility, significantly suppressing charge recombination. As a result, the resulting devices incorporating TMDS achieve a champion efficiency of 26.10% (certified 25.39%) and 24.06% at a larger area (1.0 cm$^2$) with negligible hysteresis. More importantly, the optimized devices retain 95% and 90% of their initial efficiency after 1090 h under damp heat testing (85 °C and 85% relative humidity) and after 1271 h under maximum power point-tracking conditions, respectively.

Hybrid perovskite solar cells (PSCs) have advanced rapidly over the last decade, with certified photovoltaic conversion efficiency (PCE) reaching a value of 26.7%[1–5]. Many academics are committed to promoting the industrialization of PSCs, aiming to address the increasingly severe energy crisis and environmental concerns[6,7]. Nevertheless, the integration of a typical doped hole transport layer (HTL) in traditional n-i-p structured PSCs presents significant challenges for achieving both high efficiency and operational reliability[8]. In contrast, widespread attention has been directed towards inverted PSCs (p-i-n) owing to their compatibility with the distinct bottom cells of tandem solar cells, low-temperature processability, and exceptional stability[9,10].

In inverted PSCs, fullerene derivatives such as [6,6]-phenyl-$C_{61}$-butyric acid methyl ester (PCBM) or $C_{60}$ are commonly employed as electron transport layers (ETLs)[9,11,12]. Compared to $C_{60}$, PCBM possesses a phenylbutanoate methyl ester group, which effectively reduces defects on the perovskite surface and minimizes carrier loss[9]. Simultaneously, PCBM exhibits better interface compatibility with the perovskite, which is beneficial for fabricating high-performance devices. The remarkable electron transport capability of PCBM allows for efficient electron extraction from the perovskite layer. Nonetheless, its intrinsic stability remains a critical factor influencing device performance and stability. Furthermore, PCBM molecules easily aggregate into large clusters, lowering the interfacial area required for exciton dissociation[13]. The relatively low ionization potential of PCBM causes serious charge recombination phenomena[14,15]. In addition, the energy level alignment between the lowest unoccupied molecular orbital of PCBM and the work function of the perovskite is suboptimal, reducing the device performance[16]. Although considerable efforts are being made to develop alternative ETLs to replace fullerene derivatives, their

[1]College of Optoelectronic Engineering, Chongqing University, Chongqing, China. [2]Wuhan National Laboratory for Optoelectronics, Huazhong University of Science and Technology, Wuhan, Hubei, China. [3]These authors contributed equally: Cheng Gong, Haiyun Li, Zhiyuan Xu, Yuheng Li. ✉e-mail: xiongli@hust.edu.cn; zangzg@cqu.edu.cn

performance is still significantly inferior to that of fullerene derivatives-based devices[6,17,18]. Hence, enhancing the performance and stability of fullerene derivatives is crucial for achieving high-performance inverted PSCs[19,20].

Numerous researchers are currently focused on addressing this issue. Yang et al.[21] employed a steric hindrance-assisted strategy to synthesize single isomers of $C_{60}$-and $C_{70}$-based diethylmalonate functionalized bisadducts ($C_{60}$BB and $C_{70}$BB). Simultaneously, they discovered that different solvents may efficiently adjust the molecular stacking in fullerene dimer films, resulting in dense amorphous fullerene dimer films with high electron mobility. This modification improved the performance of tin-based perovskite solar cells. Yin et al.[9] polymerized $C_{60}$ fullerene with 1, 4-bis(dodecylthio)benzene, developing an electron transport material named PFBS-$C_{12}$ polyfullerene. This material could form more conformal contact with perovskite films, leading to more effective charge collection and improved performance of blade-coated p-i-n PSCs. Bin et al.[22] designed and synthesized a series of 1, 3-dimethyl-2-phenyl-2, 3-dihydro-1H-benzoimidazole (DMBI) derivatives, which function as effective solution-phase n-type dopants, facilitating hydrogen transfer and easily forming highly active organic radicals in solution. Finally, the n-doping of PCBM by organic radicals improves the electrical properties of PCBM films. These examples highlight the effectiveness of modifying fullerene derivatives to enhance the performance of inverted PSCs. However, enhancing the electron extraction ability of PCBM remains challenging and urgently requires additional attention.

Herein, we achieved a more uniform and electron-transfer-enhanced PCBM film by incorporating tetramethylthiuram disulfide (TMDS). The surface morphology of PCBM modified by TMDS exhibits uniform distribution with almost no aggregation, contributing to the complete coverage of PCBM on the perovskite layer. Under UV light conditions, TMDS readily forms highly reactive reducing organic radicals in solution, facilitating n doping of the classical electron acceptor PCBM. TMDS-modified PCBM films can significantly suppress charge recombination by improving electron extraction. Ultimately, the champion PCE of the target device reaches 26.10% (certified 25.39%), while the $1\,cm^2$ device achieves a PCE of 24.06%. Under simulated AM 1.5 illumination, the target device maintains an initial efficiency of over 95% after 1271 h of continuous maximum power point tracking (MPPT). After 1090 h of aging at 85 °C and 85% relative humidity (RH), the encapsulated target device retains more than 90% of its initial PCE.

## Results
### Mechanism of n doping and passivation
The PSCs structure and interaction schematic between TMDS or sulfur radicals and PCBM or perovskite is illustrated in Fig. 1a. The reducing sulfur radicals generated by TMDS can transfer electrons to the carbon cage of PCBM, while the C = S bond can effectively coordinate with uncoordinated $Pb^{2+}$ on the surface of perovskite layers. The disulfide bond in the TMDS molecule is susceptible to cleavage, resulting in the generation of sulfur radicals upon exposure to UV light irradiation. This photoinduced, catalyst-free, low-temperature reaction has been demonstrated to be an effective method for generating sulfur radicals[23]. Simultaneously, PCBM serves as a typical electron acceptor that readily captures single electrons from reducing sulfur radicals to facilitate n doping (Fig. 1b)[24,25]. The products formed during the UV light illumination of TMDS can be confirmed through time-of-flight mass spectrum results (Fig. 1c). In addition to the characteristic peaks representing PCBM, there are two distinct peaks at 119.3 and 239.2, corresponding to the generated sulfur radical and TMDS monomer molecules, respectively. This confirms that disulfide bonds in TMDS break after exposure to UV light. The formation of n doping PCBM can be evidenced by electron spin resonance (ESR), with a clear paramagnetic signal observed in the PCBM with TMDS solution indicating

the production of fullerene radical anions under UV light irradiation (Fig. 1d)[22]. Furthermore, n doping can be confirmed by X-ray photoelectron spectroscopy (XPS). In Fig. 1e, after UV light irradiation, TMDS and PCBM with TMDS all exhibit three distinct chemical states of sulfur (S). The binding energy of -C-S-S-C- and -C = S in these two samples show no significant difference, while the binding energy of -C-S differs by 0.15 eV. The increase in binding energy observed for the sulfur atom within the C-S bond in the PCBM with TMDS sample suggests a decrease in the electron cloud density surrounding sulfur, thus confirming the electron transfer between sulfur radicals and PCBM. After UV light irradiation, characteristic peaks of the S-S and C = S functional groups are still observed in the PCBM with the TMDS sample (Supplementary Fig. 1), indicating that unreacted TMDS monomers remain after UV light irradiation. In addition, a new peak appears at 839.8 cm$^{-1}$ in the FTIR spectrum of the PCBM with TMDS sample (Supplementary Fig. 2a). Similarly, in the UV–vis absorption spectra, the PCBM with TMDS sample exhibits a new peak at 480 nm (Supplementary Fig. 2b). Combining the results of FTIR and UV–vis absorption spectra, it is suggested that after the charge transfer process, the organic radicals may become positively charged dipoles upon losing electrons, while PCBM becomes negatively charged dipoles upon gaining electrons. Therefore, this promotes the formation of dipole-dipole interactions between them, which helps to stabilize the structure of the complex[26].

In addition, the -C = S functional group present in TMDS strongly coordinates with uncoordinated $Pb^{2+}$. We solely coated the TMDS material on the surface of the perovskite and characterized the samples using XPS. It can be seen that the binding energy of sulfur in C = S shifts from 161.33 to 161.72 eV, while the binding energy of sulfur in -C-S remains almost unchanged (Fig. 1f). This indicates that the -C = S bond in TMDS can strongly interact with perovskite components. Correspondingly, the characteristic peak of Pb 4$f$ also shifts toward lower binding energy by 0.45 eV, suggesting effective coordination between uncoordinated $Pb^{2+}$ on the perovskite surface and -C = S in TMDS (Fig. 1g). Furthermore, the vibrational characteristic peaks of -C = S in the FTIR spectrum show a noticeable shift (Fig. 1h), consistent with the results obtained from XPS. In summary, these findings demonstrate that TMDS can generate sulfur radicals when exposed to UV light, thus forming n doping effect on PCBM. In addition, the TMDS monomer itself containing C = S constructs a stable binding with uncoordinated $Pb^{2+}$ in perovskite.

### Impact on the PCBM layer
The morphology and electrical properties of PCBM modified by TMDS are subsequently investigated. A significant reduction is found in the root-mean-square (RMS) roughness of the film, from 2.35 nm in pristine PCBM film to as low as 1.52 nm in PCBM with TMDS one, as shown in Fig. 2a–c. The enhanced smooth surface facilitates optimal contact with the upper electrodes, thereby improving carrier transfer. The PCBM material is widely used as an organic semiconductor in various applications. However, its tendency to form large clusters hinders the complete coverage of the perovskite layer and reduces the interfacial area available for exciton dissociation. As shown in Supplementary Fig. 3a, b, the pristine PCBM solution exhibits a diverse range of particle sizes, with diameters exceeding 10 nm. However, upon the addition of TMDS, the particle size distribution of the PCBM solution becomes notably more uniform. The uniform particle size is attributed to the strong interaction between TMDS and PCBM, hindering the aggregation of the solution beyond individual unit particles[27]. To investigate the surface morphology of PCBM deposited onto perovskite thin films, PCBM with or without TMDS was spin-coated onto the surfaces of perovskite films prepared under identical conditions. Figure 2d, e and Supplementary Fig. 4a–d clearly illustrate the presence of numerous holes on the surface of a PCBM film without TMDS, which compromises effective electron extraction and aggravates charge recombination. In contrast, when TMDS is introduced to PCBM, remarkable improvements are observed in terms of the uniformity of

the PCBM films, with the complete elimination of holes. After comparison, it was found that the weak interaction between PCBM and perovskite films led to the formation of more pores in the control PCBM film. The target PCBM forms more conformal contact with the perovskite due to the coordinating effect between its TMDS and uncoordinated $Pb^{2+}$ on the perovskite surface, resulting in a more

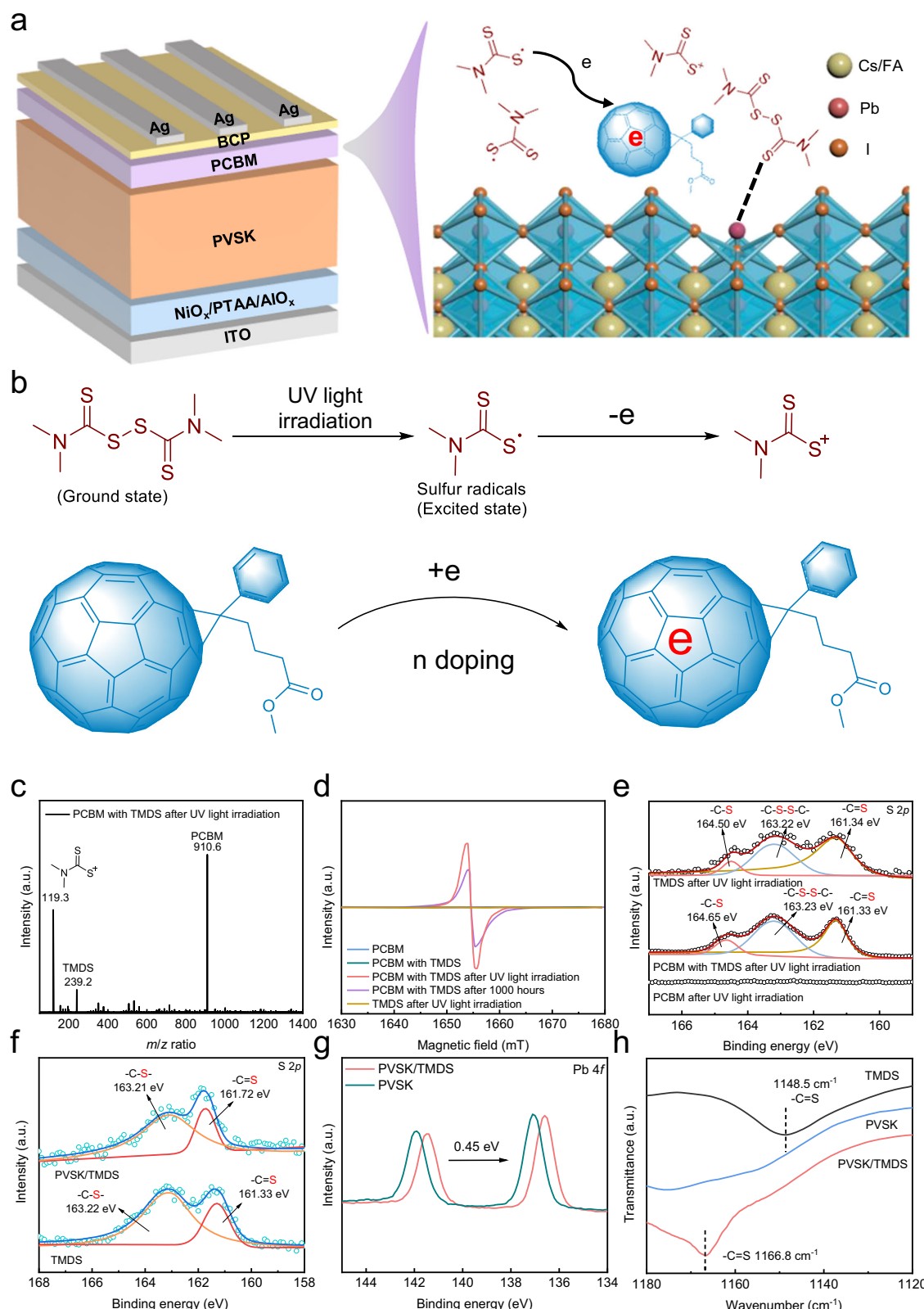

**Fig. 1 | N doping effect and interactions. a** PSCs structure and schematic illustration of chemical interactions between perovskite and TMDS or sulfur radicals. **b** Formation of sulfur radicals and n doping mechanism of PCBM. **c** Time-of-flight mass spectrum of PCBM with TMDS film. **d** The ESR spectra of different solutions. **e** XPS spectra of S 2*p* for TMDS and PCBM films without and with TMDS. XPS spectra of (**f**) S 2*p* and (**g**) Pb 4*f*, for TMDS and PVSK films without and with TMDS. **h** FTIR spectra of TMDS and PVSK films without and with TMDS.

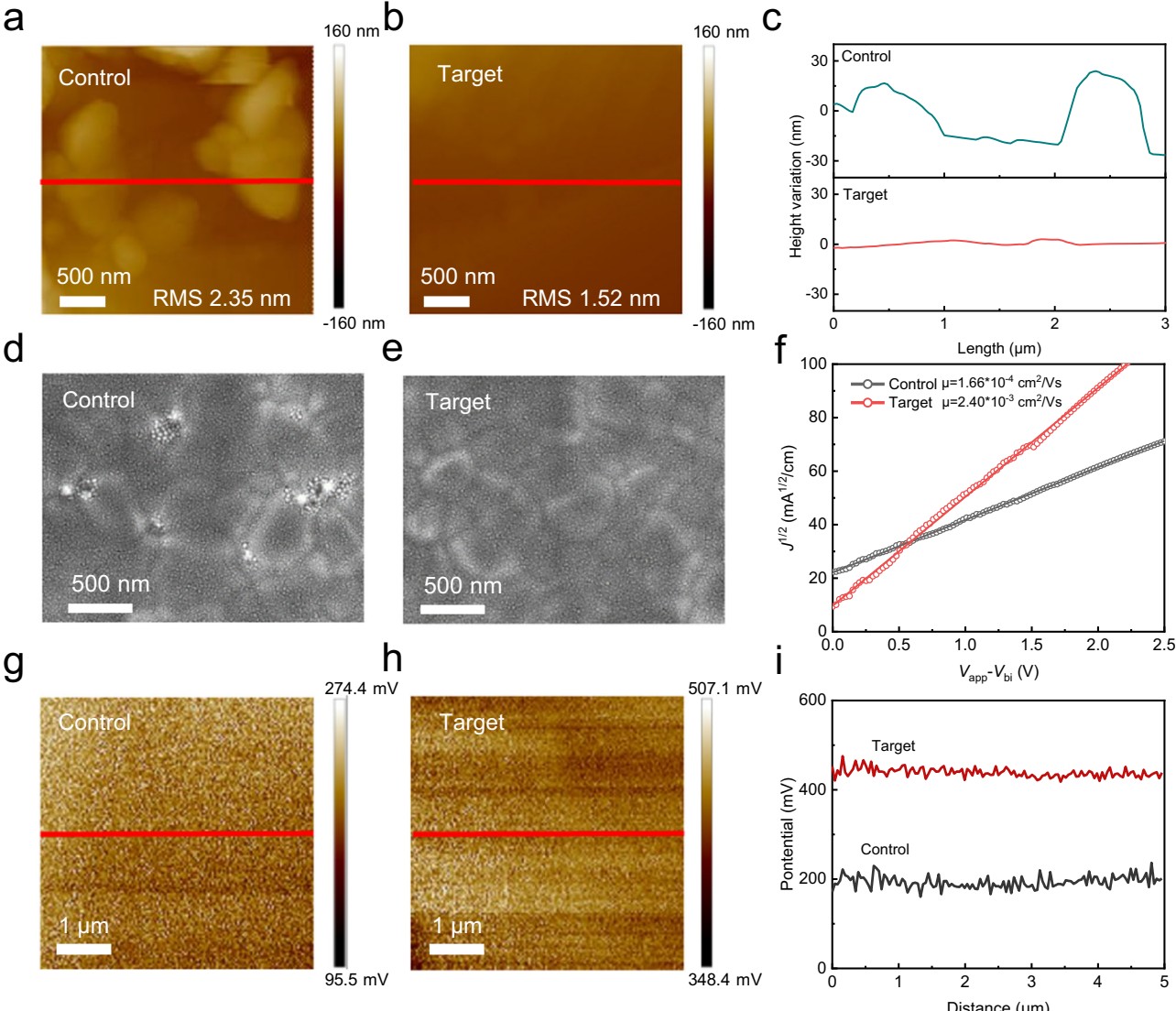

**Fig. 2 | Morphology and electrical properties. a, b** AFM topography images and the (**c**) corresponding line profiles for PCBM films with and without TMDS treatment. **d, e** High-resolution top-view SEM images for PVSK/PCBM with and without TMDS. **f** The mobility of PCBM films with and without TMDS treatment. **g, h** KPFM surface potential images and the (**i**) corresponding line profiles for PCBM films with and without TMDS treatment.

uniform morphology of the PCBM film[9]. In addition, the ToF-SIMS spectrum also shows that TMDS can diffuse into the perovskite phase or the grain boundaries, displaying a gradient distribution (Supplementary Fig. 5). This facilitates the deep passivation of TMDS within the perovskite layer.

The current-voltage (I–V) test results for ITO/ETL/Ag structured devices are presented in Supplementary Fig. 6. As the concentration of TMDS increases, the conductivity of PCBM with TMDS films exhibits a trend of initially rising and then declining. When the TMDS concentration reaches $2 \times 10^{-3}$ mmol/ml, the conductivity reaches its maximum value of $1.22 \times 10^{-5}$ S/cm, while the pristine PCBM only reaches $3.45 \times 10^{-6}$ S/cm. In addition, to evaluate the electron mobility of different electron transport layers by using the space charge-limited current (SCLC) method, we fabricated ITO/SnO$_2$/PCBM (W/WO TMDS)/SnO$_2$/Ag structures, as depicted in Fig. 2 f. It is observed that PCBM with TMDS exhibits enhanced electron mobility ($2.4 \times 10^{-3}$ cm$^2$/V S) compared to that of pristine PCBM ($1.66 \times 10^{-4}$ cm$^2$/V S). This significant improvement in both electron mobility and conductivity can be attributed to the n-doping effect of sulfur radicals on PCBM, resulting in an increased concentration of electrons within the PCBM film. Elevated electron concentration

augments participation in conduction, thereby enhancing film conductivity. Electron mobility is mainly affected by the scattering of lattice defects or impurity atoms. As electron concentration increases, the average distance between electrons decreases. Although the probability of electron-electron collision scattering increases, the probability of electron scattering by lattice defects or impurity atoms significantly decreases[28–30]. As a result, the overall charge carrier mobility demonstrates enhancement. Thus, the n doping effect enables enhanced electron transfer in the fullerene ETL. The results depicted in Fig. 2g–i from kelvin probe force microscopy (KPFM) demonstrate that the surface potential of the PCBM with TMDS film is consistently greater than that of the control PCBM film[31]. Moreover, the surface is more uniform, thereby impeding charge recombination and facilitating enhanced charge transfer from the perovskite film.

## Defect passivation and carrier transport
The ultraviolet photoelectron spectroscopy (UPS) (Fig. 3a, Supplementary Fig. 7a–f, and Supplementary Table 1) was used to investigate the impact of TMDS on the energy level alignment. The conduction band minimum (CBM) of PCBM undergoes a shift from −4.31 to

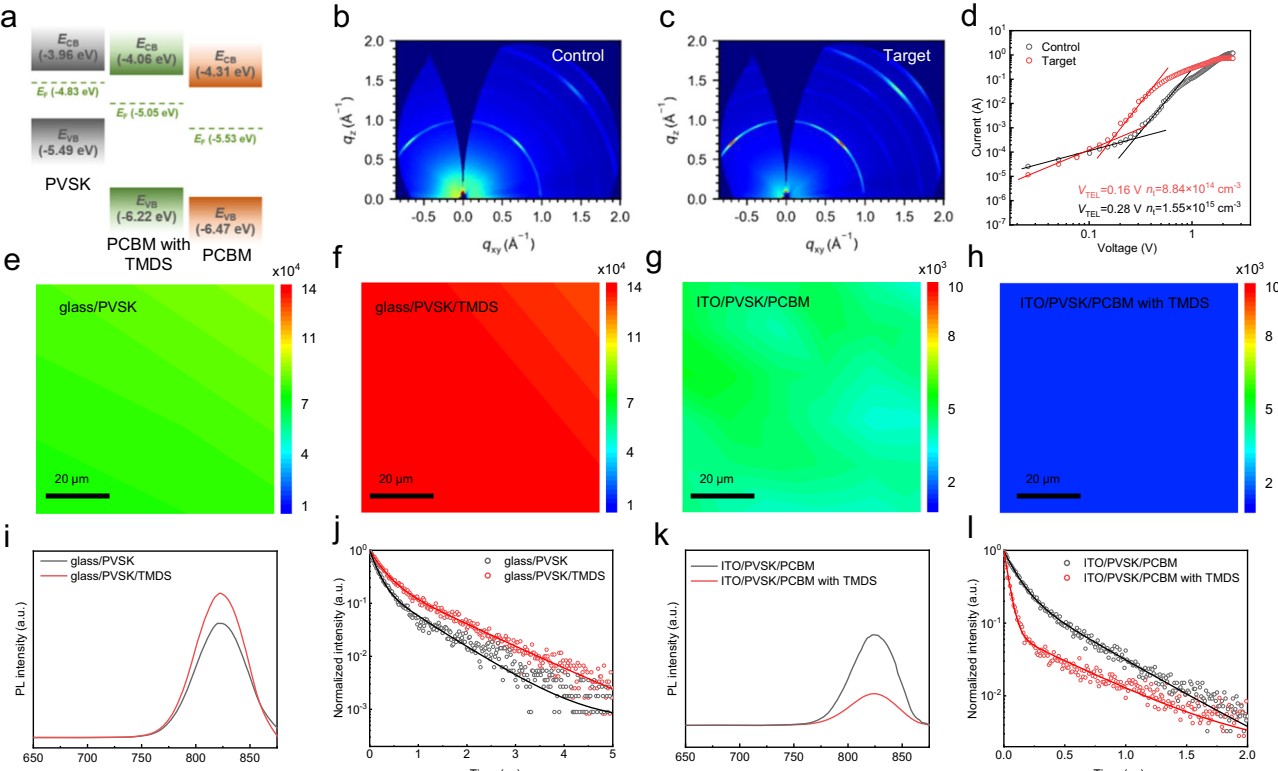

**Fig. 3 | Passivation and carrier transport. a** Energy level diagram of the PCBM films without and with modifiers as well as perovskite film. **b, c** GIWAXS mappings of the perovskite films with and without TMDS treatment. **d** SCLC plots of the electron-only device ITO/SnO₂/perovskite/PCBM/Ag where the PCBM films without and with TMDS were used. **e, f** PL mapping images of glass/perovskite without and with TMDS. **g, h** PL mapping images of the glass/perovskite/PCBM without and with TMDS. **i** PL and (**j**) TRPL spectra of the glass/perovskite without and with TMDS. **k** PL and (**l**) TRPL spectra of the glass/perovskite/PCBM without and with TMDS.

−4.06 eV, while the Fermi level experiences a change from −5.53 to −5.05 eV due to the n doping effect induced by sulfur radicals. This significant reduction in the CBM energy offset between PCBM and perovskite films facilitates efficient electron extraction and minimizes non-radiative recombination losses at the interface. TMDS can passivate uncoordinated $Pb^{2+}$ defects on the surface of perovskite, contributing to an enhancement in the crystallinity of the perovskite film. The GIWAXS results indicate that the surface of the perovskite modified with TMDS exhibits Bragg spots with higher intensity and sharper features compared to the pristine perovskite (Fig. 3b, c). This is also evident from the XRD results, where the characteristic diffraction peak intensity of the perovskite film surface modified by TMDS is significantly greater than of the pristine perovskite (Supplementary Fig. 8). The diffraction peaks of the target perovskite films at (001) and (002) planes exhibit a reduced full width at half maximum (FWHM), indicating a higher quality of the target perovskite films. We quantified the trap density of the pristine perovskite and TMDS-surface-modified perovskite films by the space charge limited current (SCLC) measurement for electron-only devices[32,33], as shown in Fig. 3d. The trap density of the TMDS-modified perovskite film is minimal, as low as $8.84 \times 10^{14}$ cm$^{-3}$, while the trap density for the control film increased to $1.55 \times 10^{15}$ cm$^{-3}$.

Similarly, the pristine perovskite surface exhibits weak fluorescence intensity. After TMDS treatment, the perovskite film demonstrates strong fluorescence signals, indicating an extended carrier lifetime (Fig. 3e, f, i, and j and Supplementary Table 2). The strong interaction between C = S and $Pb^{2+}$ decreases the defect density of the perovskite film. For the ITO/PVSK/PCBM with/without TMDS samples, the electron quenching in the control sample shows severe hysteresis (Fig. 3g, h). As mentioned earlier, these issues arise from the limited ability of pristine PCBM to extract electrons, as well as the overall non-uniformity of the film. Clearly, the TMDS-modified samples show rapid electron

quenching and good uniformity, indicating a significant increase in the ability to extract electrons and a decrease in the number of charge recombination centers. The decrease in the PL intensity and shortened carrier lifetime also confirm the optimized extraction of charges (Fig. 3k, l, and Supplementary Table 3). Fits to the TRPL transients were used to compute the differential lifetime using Krogmeier et al.'s model[34–36] (Supplementary Fig. 9). The charge transfer process at early times (~ 250 ns) led to a faster rise of differential lifetime in target film than that of control film. The transition from increasing lifetime to the plateau marks the end of charge transfer, and non-radiative first-order recombination becomes dominant. The TRPL measurements have been conducted at multiple excitation intensities. As shown in Supplementary Fig. 10, the carrier lifetime was increased after incorporation of TMDS, and the same tendency was observed at different excitation intensities. However, it is worth noting that the charge carrier lifetime was also reduced slightly with the increase of excitation intensity due to the increase of bimolecular recombination losses[37]. Meanwhile, the variation of carrier lifetime of target samples depending on excitation intensity was not as significant as that of the control sample, which is indicative of the lower defect density in the target samples.

The conclusion was further validated by the results depicted in Supplementary Fig. 11 of the Supporting Information, which demonstrated the dependency of open-circuit voltage ($V_{OC}$) and short-circuit current density ($J_{SC}$) on light intensity. The ideal factor for the control PSCs is 1.34, while the ideal factor for TMDS-modified PSCs is 1.20. This indicates an enhancement in charge transfer and transport, as well as the suppression of non-radiative recombination channels. In Supplementary Fig. 12, devices based on TMDS exhibit a higher built-in potential ($V_{bi}$) than that of the control device. This suggests that the enhanced carrier separation and extraction are attributed to a more favorable band alignment. The TMDS-modified device exhibited a

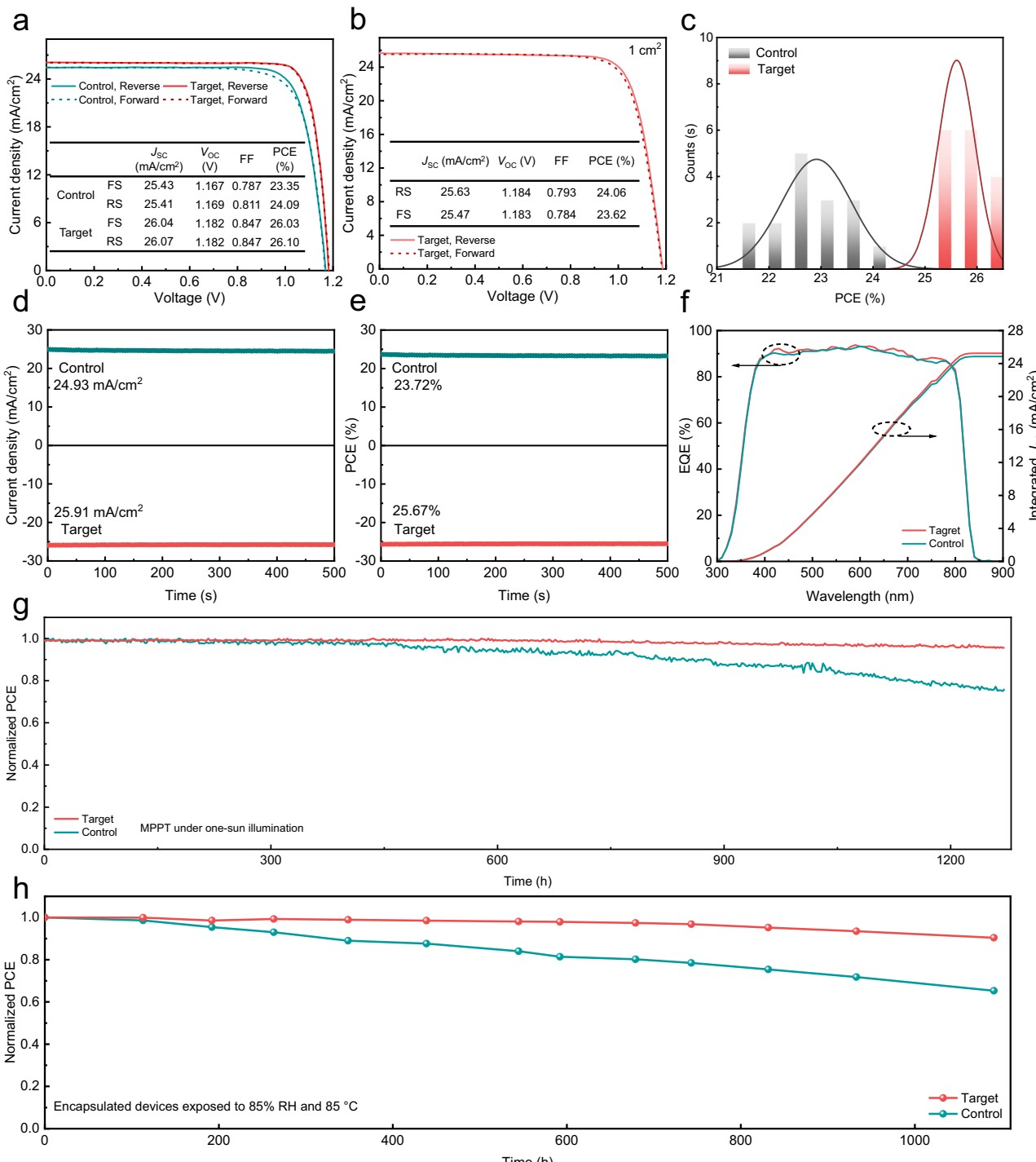

**Fig. 4 | Photovoltaic Performance. a** Forward and reverse scans of both the champion control and target PSCs were conducted to generate *J-V* curves, with relevant photovoltaic parameters depicted in the insets. **b** The *J–V* characteristics of the highest-performing target device, having a 1 cm² surface area. **c** PCE histograms for PSCs with and without TMDS. Steady-state (**d**) current density and (**e**) PCE versus time for the best-performing devices measured at the maximum power point. **f** EQE spectra of the PSCs without and with TMDS. **g** The PCE of both the unencapsulated control devices and those modified with TMDS was assessed at the MPPT under sustained one-sun exposure within a chamber environment maintained at approximately 40 °C. **h** PCE of encapsulated devices accelerated aging at 85 °C and 85% relative humidity.

notable swifter decay in photocurrent (0.21 µs) compared to the control device (0.86 µs) (Supplementary Fig. 13a). In addition, TMDS-modified devices exhibit a slower decay rate of the photovoltage (2.88 µs), contrasting sharply with the rapid decay rate of the control devices (1.67 µs) (Supplementary Fig. 13b). Therefore, after the addition of TMDS, the transport and extraction of charge carriers become more efficient.

## Photovoltaic performance and long-term stability

Figure 4a shows the *J-V* characteristics of the optimal control and target devices. The control device demonstrates a PCE of 24.09%, a $J_{SC}$ of 25.41 mA/cm², a $V_{OC}$ of 1.169 V, and a fill factor (FF) of 81.1%. In comparison to the control device, the target device demonstrates a PCE of 26.10%, $J_{SC}$ of 26.07 mA/cm², $V_{OC}$ of 1.182 V, and FF of 84.7%. The certified PCE was determined to be 25.39% with a $J_{SC}$ of 25.78 mA/cm², a

$V_{OC}$ of 1.179 V, and an FF of 83.55% (Supplementary Fig. 14a–c). Furthermore, the 1 cm$^2$ device modified with TMDS demonstrates a PCE of 24.06%, a $J_{SC}$ of 25.63 mA/cm$^2$, a $V_{OC}$ of 1.184 V, and a FF of 79.3% (Fig. 4b). We compared the photovoltaic performance of the control and target devices (Supplementary Fig. 15 and Supplementary Table 4). Compared to the control devices with an average PCE of 22.91 ± 0.67%, the average PCE of the target devices increased significantly to 25.60 ± 0.35%, mostly due to the enhancements in $V_{OC}$ and FF (Fig. 4c). We evaluate the compatibility of the TMDS additive with upscaling processes by fabricating perovskite modules on a 30 cm × 30 cm substrate. As shown in Supplementary Fig. 16, the active area of the module is 669 cm$^2$ in this context. The PCE of the champion module is 17.05% and 15.30% for target and control devices, respectively. The improved photovoltaic performance is primarily due to the effective passivation of $Pb^{2+}$ defects on the surface of the perovskite, as well as the more ideal alignment of the energy levels between the PCBM transport layer and the perovskite layer, which accelerates the extraction and transport of charge carriers.

Similarly, the steady-state output of the target device improves significantly in both the PCE and $J_{SC}$, reaching 25.67% and 25.91 mA/cm$^2$ ($V_{max}$ = 0.99 V), respectively, compared to the control values of 23.73% and 24.93 mA/cm$^2$ ($V_{max}$ = 0.95 V). This improvement is consistent with the $J$-$V$ results (Fig. 4d, e). Figure 4f shows that after TMDS modification, the device exhibits a significant increase in the external quantum efficiency (EQE) in the wavelength range of 400–800 nm, resulting in the integrated current density of the target device reaching 25.25 mA/cm$^2$, while the integrated current density of the control device is only 24.16 mA/cm$^2$. It is important to note that the slight mismatch in the spectra between the IPCE light source and the solar simulator leads to minimal differences in the EQE and $J$-$V$ curves[38]. As anticipated, the TMDS-modified devices outperform the control devices in terms of photovoltaic performance. This difference is attributed to the n doping effect of TMDS on PCBM, improving the alignment of energy bands, reducing losses caused by non-radiative recombination, and thereby enhancing the transport and extraction of charge carriers.

Finally, we evaluated the long-term stability of both the unencapsulated control and target devices when subjected to MPPT under continuous one-sun irradiation (45 °C). After TMDS modification, the device maintained 95.6% of its initial efficiency after 1271 h, whereas the control device only retained 76.0% (Fig. 4g). The initial PCE of the control and TMDS-modified devices were 23.43% and 25.63%, respectively. Following accelerated aging at 85 °C and 85% relative humidity for 1090 h, the encapsulated target and control devices retained 90.4% and 65.3% of their initial efficiency, respectively (Fig. 4h). The initial efficiencies for the control and target devices were 23.57% and 25.49%, respectively.

## Discussion

In summary, we propose a strategy to enhance the electrical properties and surface morphology of PCBM films by introducing TMDS into PCBM solution to generate reducing radicals under UV irradiation conditions. The electron acceptor PCBM absorbs the free electrons from these sulfur radicals, leading to the n doping effect in PCBM. As a result, TMDS-modified PCBM films not only exhibit higher electrical conductivity and electron mobility, allowing for more efficient extraction and transportation of electrons from the perovskite layer, but they also possess better energy band alignment with the perovskite layer, minimizing charge recombination losses at the interface. In addition, TMDS molecules or sulfur radicals within PCBM can bind with uncoordinated $Pb^{2+}$ on the surface of perovskite, effectively reducing surface defects. Finally, this approach yielded a device efficiency of 26.10% (certified 25.39%) without hysteresis for our target device. Moreover, a 1 cm$^2$ device achieved an efficiency of 24.06%. More importantly, the target device maintained over 95 and 90.4% of its initial efficiency even after 1271 h of MPPT and 1090 h at a temperature of 85 °C and RH of 85%, respectively.

## Methods

### Materials

The lead (II) iodide ($PbI_2$, 99.99%), cesium iodide (CsI, 99.99%), [6,6]-phenyl $C_{61}$ butyric acid methyl ester ($PC_{61}BM$), and formamidine hydroiodide (FAI, 99.5%) were purchased from Advanced Election Technology Co., Ltd. Nickel nitrate hexahydrate ($Ni(NO_3)_2 \cdot 6H_2O$, 99.999%), sodium hydroxide (NaOH, 99.9%), N,N-dimethylformamide (DMF, 99.8%), isopropanol (IPA, 99.5%), dimethyl sulfoxide (DMSO, 99.9%), chlorobenzene (CB, 99.8%), and $Al_2O_3$ dispersed solution in IPA with a concentration of 20 wt.% were obtained from Sigma Aldrich. Poly (triaryl amine) (PTAA) with a molecular weight distribution of 6000–15000, phenethylamine hydroiodide (PEAI, 99.5%) and bathocuproine (BCP) were purchased from Xi'an Polymer Light Technology Corp., while the chemicals including: tetramethylthiuram disulfide (TMDS, 97%) were bought from Aladdin. The $NiO_x$ nanoparticles (NPs) were synthesized based on previous research[39]. All chemicals and solvents used in this study were utilized without any additional purification.

### Device fabrication

The ITO-coated glass substrates were laser-etched, followed by ultrasonic cleaning of the etched ITO glass for 15 min using a detergent, deionized water, ethanol, and isopropanol in sequential order. The ITO-coated glass substrates were subjected to a 20-minute treatment of ultraviolet ozone (UVO). Subsequently, a $NiO_x$ NPs aqueous ink with a concentration of 25 mg/mL was prepared by dispersing the as-prepared $NiO_x$ NPs in deionized water. This ink was then spin-coated onto the ITO glass at a speed of 5000 rpm for 30 s. The $NiO_x$ films were annealed at 150 °C for 10 min, followed by immediate transfer into a nitrogen-filled glove box. The $NiO_x$ films were spin-coated with a 2 mg/mL PTAA solution in CB at 6000 rpm for 30 s. Subsequently, the PTAA films were spin-coated with an $Al_2O_3$ dispersion solution (0.4 wt% in IPA) at 5000 rpm for 30 s. The $FA_{0.95}Cs_{0.05}PbI_3$ perovskite film was prepared by dissolving 228.4 mg of FAI, 18.2 mg of CsI, 645.4 mg of $PbI_2$ and 1 mg of PEAI in a mixed solvent solution (v/v, DMF: DMSO = 4: 1) with a concentration of 1.4 mmol/mL. The perovskite precursor solution was then spin-coated onto a glass/ITO/$NiO_x$/PTAA/$Al_2O_3$ substrate at speeds of 2000 rpm for 10 s, followed by an additional spin at 4000 rpm for 40 s. During the second spin coating step, 150 μL of CB was deposited onto the perovskite film 5 seconds before the program ended. The resulting wet perovskite films were annealed at 100 °C for 30 min. Subsequently, a solution of $PC_{61}BM$ in CB with a concentration of 23 mg/mL was spin-coated onto the perovskite films at a speed of 2500 rpm for 40 s. For the modified PCBM layer, TMDS with different concentrations was added to the $PC_{61}BM$ solution for modification, followed by UV light irradiation for 2 h. 5 mg BCP was added into 1 mL IPA to prepare a supersaturated solution, which was filtered by a PTFE filter before use. Afterward, the obtained saturated solution was spin-coated on PCBM film at 5000 rpm for 30 s. Finally, a thermal evaporation process under vacuum conditions ($2 \times 10^5$ Pa) was employed to deposit an Ag electrode with a thickness around 100 nm.

For the modules, first, use Physical Vapor Deposition (PVD) technology to prepare $NiO_x$ with a thickness of approximately 18 nm on the FTO substrate. The $FA_{0.83}Cs_{0.17}PbI_3$ perovskite film is prepared by dissolving 0.2086 g of $PbCl_2$, 1.325 g of CsI, 0.3038 g of MACl, 15.213 g of $PbI_2$, and 4.282 g of FAI in a 30 ml mixed solvent solution (v/v, DMF: NMP: ACN = 6: 1: 3) with a concentration of 1.1 mmol/mL. The perovskite precursor solution is then slot-die coated onto a glass/FTO/$NiO_x$ substrate with a slot-die gap of 100 μm above the substrate at a speed of 5 mm/s. Then, the resulting wet perovskite films are quickly transferred into a vacuum chamber, which is pumped to 10 Pa and maintained for 40 s. The resulting perovskite films are then annealed at 150 °C for 15 min in the air (RH = 20%). Subsequently, the PCBM solution (23 mg/mL) is then slot-die coated onto the perovskite

film with a slot-die gap of 100 μm at a speed of 4 mm/s. 20 nm of ALD-SnO$_2$ is deposited on the PCBM surface. Finally, the ITO/Cu/ITO sandwich electrodes are prepared on the ALD-SnO$_2$ surface using PVD technology, with thicknesses of 20 nm/15 nm/20 nm, respectively.

## Characterization of the solar cells

The $J$–$V$ parameters of the devices were assessed in ambient air (with a relative humidity of 40–50%) using an AM 1.5 G solar simulator equipped with a 450 W xenon lamp (Newport-2612A) and a Keithley 2400 Source Meter. Light intensity was adjusted to AM 1.5 G one sun (100 mW/cm) through a NIM-calibrated standard Si solar cell. The active area of the cells was defined as either 0.09 cm$^2$ or 1 cm$^2$ using a black metal mask. For EQE measurement, a Newport Instruments system (Newport-74125) coupled with a lock-in amplifier and a 300 W xenon lamp was utilized. Transient photocurrent and photovoltage measurements were conducted using a system excited by a 532 nm (1000 Hz, 6 ns) pulsed laser. The recording of the photocurrent or photovoltage decay process employed a 1 GHz Agilent digital oscilloscope (DSOX3102A) with a 50 X or 1 MX sampling resistor.

## Characterization of the device stability

The stability of unencapsulated devices was assessed under continuous simulated solar illumination equivalent to one sun, emitted by a light-emitting diode (LED) lamp at 45 °C without a UV filter, at a controlled temperature of 25 ± 5 °C. MPPT was executed within a nitrogen-filled glovebox utilizing a dedicated MPPT system (YH-VMPP-S-16). Following the ISOS-T-1 standard, encapsulated devices were subjected to accelerated aging tests at 85 °C and 85% relative humidity within a specialized climate chamber (ZK-301). Encapsulation was performed in a nitrogen atmosphere, involving the sealing of the device with a top glass cover and edge-sealing using a UV-curable adhesive. Precise alignment of the top glass with the device's perimeter was achieved using tweezers, followed by the application of ~ 200 μl of UV adhesive into the intervening gap. Curing of the adhesive was then accomplished through a two-minute exposure to ultraviolet light.

## Other Measurements

The PCBM film's surface morphology was measured using field emission scanning electron microscopy (Apreo S HiVac FEI). X-ray diffraction patterns were obtained with a PANalytical Empyrean diffractometer equipped with Cu Kα radiation ($\lambda$ = 1.5406 Å). Recording photocurrent or photovoltage decay process used a 1 GHz Agilent digital oscilloscope (DSOX3102A) with a 50 X or 1 MX sampling resistor. Mott-Schottky measurements (1000 Hz) were conducted on a Chenhua electrochemical workstation (CHI 760E), and frequency-dependent capacitance measurements were performed on the same workstation. Steady-state PL spectra and TRPL spectra were obtained by a fluorescence spectrophotometer (FLS1000). Raman mapping (LabRAM HR Evolution) was recorded under 532 nm excitation. UPS and XPS measurements were conducted using a monochromatized Al source (Escalab Xi + ). XPS was calibrated using the peak position of C 1 s, and UPS was calibrated using the work function of Au. FTIR spectra were measured by Nicolet iS50 Infrared Fourier spectrometer. KPFM measurements were performed using a Bruker Dimension Icon (Germany) with AFM conducting tips featuring a resonance frequency ($\omega_0$) of ~ 140 kHz and a spring constant of 5.0 N/m. Standard AC mode imaging was employed for topography acquisition in the KPFM measurement. Short-circuit conditions were established by directly connecting the Ag electrode and the ITO electrode. The ESR measurements had been performed on a Bruker EMXnano band pulsed-ESR spectrometer. A standard sample of the magnetic field (sodium thiosulfate pentahydrate) was used to calibrate the magnetic field. For the PCBM solution, 23 mg of PCBM was added to 1 mL of CB solution; for the PCBM modified by TMDS solution, 23 mg of PCBM

and 0.48 mg ($3 \times 10^{-3}$ mmol/mL) of TMDS were added to 1 mL of CB solution. Samples requiring UV light treatment were irradiated with 365 nm ultraviolet light for 2 h.

## Reporting summary

Further information on research design is available in the Nature Portfolio Reporting Summary linked to this article.

## Data availability

Source data are provided in this paper. All the data supporting the findings of this study are available within this article and its Supplementary Information. Any additional information can be obtained from corresponding authors upon request. Source data are provided in this paper.

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

## Acknowledgements

This work was financially supported by the Fundamental Research Funds for the Central Universities (2022CDJQY-010), the National Natural Science Foundation of China (62204028, 11974063), Natural Science Foundation of Chongqing (CSTB2022NSCQ-MSX1514), China Postdoctoral Science Foundation (2022TQ0391, 2022M710507) and Chongqing Postdoctoral Science Special Foundation (2022CQBSHTB1026). We would like to thank the Analytical and Testing Center of Chongqing University for various measurements.

## Author contributions

C.G. conceived the idea and designed experiments. C.G., B.W., A.W., Z.G., Q.Z., and C.Z. conducted the experiments and prepared films and devices. Z.Z., Z.G., Z.L., H.W., and C.G. designed and carried out the spectroscopy investigations and data analysis. C.G., Y.L., AW., and H.L. characterized thin film and devices. C.G., Y.L., and H.L. fabricated the devices and performed certification. C.G. and Z.X. prepared the first draft of the manuscript. C.G., X.L., and Z.Z. wrote the final version of the manuscript. All authors discussed the results and reviewed the manuscript. Z.Z. supervised this project.

## Competing interests

The authors declare no competing interests.
