## [Peer Review file · Nature Communications]

Efficient and stable inverted perovskite solar cells enabled by homogenized PCBM with enhanced electron transport

Corresponding Author: Professor Zhigang Zang

Version 0:

Reviewer comments:

Reviewer #1

(Remarks to the Author)

In this manuscript, the authors reported a strategy to enhance the electrical properties and surface morphology of PCBM films by introducing TMDS into PCBM solution to generate reducing radicals under UV irradiation condition. The electron acceptor PCBM absorbed the free electrons from these sulfur radicals, leading to the n doping effect in PCBM. As a result, this approach yielded a device efficiency of 26.10% (certified 25.56%) without hysteresis for the target device. Moreover, a 1 cm² device achieved an efficiency of 24.06%. More importantly, the target device maintained over 95% and 90.4% of its initial efficiency even after 1,271 h of MPPT and 1,090 h at a temperature of 85 °C and RH of 85%, respectively. This manuscript can be accepted for publication after addressing the following issues.

1. It would be more appropriate to change the title to "Highly efficient and stable inverted perovskite solar cells enabled by homogenized PCBM with enhanced electron transport" to specify the used fullerene derivative.
2. The use of the "@" in "PCBM@TMDS" is incorrect and should be modified.
3. In the device fabrication, a solution of BCP in IPA with a concentration of 5 mg/mL was used, which should to be checked, because the concentration in most cases in the literature was 0.5 mg/mL.
4. The related references of "Energy Environ. Sci., 2016, 9, 3424; Adv. Energy Mater. 2023, 13, 2302191; Nat. Commun. 2023, 14, 8052; Adv. Mater. 2024, 36, 2308706." can be cited.

Reviewer #2

(Remarks to the Author)

The authors conducted investigations to reveal the importance of the method to improve the properties of PCBM using TMDS in inverted PSCs. Through TMDS treatment, the formation of reducing sulfur radicals could be introduced into the PCBM to achieve n-type doping, which improves its electrical conductivity and electron mobility. The results show that the TMDS-treated PCBM layer exhibits excellent properties, resulting in the TMDS device with superior photovoltaic performance. This work is of interest to the research community of inverted PSCs, which may be considered for publication if the following issues can be addressed:

1. In the introductory section, the authors claim that "C60 is insoluble in many solvents and requires expensive thermal evaporation procedures for processing, restricting its application in all-solution-processed PSCs. This issue has led to the widespread utilization of PCBM". I think this statement is wrong, the material cost of C60 is much lower than PCBM in the current trend of large-area module preparation. The C60 vapour deposition method has already reached the industrial consensus, which can help the PSCs devices to achieve large-area module preparation. Therefore, the authors need to further elaborate on the significance of modifying PCBM rather than C60.
2. The methodology for specific sampling process and electron spin resonance (ESR) measurement are not described.
3. The authors demonstrated the interaction between TMDS and PCBM by XPS, and I am curious whether TMDS would diffuse into the phase or the grain boundaries, which is also important for PSCs. So, I would suggest the authors try TOF-SIMS, which may help better characterize the spatial distribution of TMDS.
4. The authors mention on page 8 that "the homogeneous particle size is attributed to the strong interaction between TMDS and PCBM, which prevents the aggregation of more than a single unit of particles in solution." What type of interaction between TMDS and PCBM occurs? There seems to be a lack of corresponding evidence. Please provide some additional evidence such as FTIR or DFT calculations.

5. The PSCs fabricated by the spin coating method have achieved an efficiency of over 24% on devices with an area of 1 cm². However, the spin coating process is inherently not scalable. If possible, the authors should demonstrate the compatibility of this method with scalable fabrication techniques, such as blade coating, slot-die coating, spray coating, etc. We are also curious about the performance of TMDS-PCBM in large-area module devices.
6. It is strongly recommended the authors use a more complete physical model for the TRPL to demonstrate optimized extraction of charges (e.g. computed differential lifetimes from fits to the transients). The full decay is governed by nonradiative, trap-assisted surface/bulk recombination (mostly monoexponential decay), radiative recombination ("bimolecular," second-order decay), and charge transfer effects. In this manuscript, the introduction of TMDS also may cause the passivation of defects, which also change decay lifetimes. Consequently, the above analysis of the PL mapping results and the TRPL lifetime results appears to be inadequate. It is recommended that the authors incorporate light intensity-dependent PLQY or hyperspectral techniques to account for the above. It is strongly recommended the authors cite some important relevant literature (e.g. Science 370,1300-1309 (2020), Matter 7, 1054–1070 (2024)).
7. Some typographic errors should be corrected, such as Supplementary Fig. 7 and caption, lines 151.

Reviewer #3

(Remarks to the Author)

The communication entitled "Highly efficient and stable inverted perovskite solar cells enabled by homogenized fullerene with enhanced electron transport" written by Cheng Gong, et. al. has focused on introducing tetramethylthiuram disulfide (TMDS) into PCBM. Under UV light highly reducing organic materials produced in solution which helped to suppress charge recombination and increase conductivity and electron mobility, and performance of inverted perovskite solar cells compare to classical electron acceptor PCBM. While the work interesting, some of the core claims of the article are lacking sufficient justification to recommend publication.

While the TMDS generates radicals and transfers at least some of them to PCBM, it is not clear that this "doping" strategy results in long lasting effects for the PCBM. If the result is actually doping, then the radicals that are generated should be detectable on a long-term time scale. Are PCBM radicals detectable by EPR after 1000 hr, which is a similar time for the solar cell stability tests?

Figure 1d: Please provide ESR of the solution containing only the TMDS after UV radiation. Then it could be easier to conclude if the paramagnetic signal is from fullerene radical or any other radical due to radical reactions in TMDS.

Figure 1e: the authors claimed that S-S bond dissociation happens under UV irradiation. I wonder why we can still see S-S bond curve even with more intensity after irradiation? While Fig 1S also show only a weak peak in FTIR compare to the TMDS. Further, there is not obviously a significant difference between the data for TMDS+UV and TMDS+PCBM+UV. The data are not sufficiently clean to make the claims that the authors are making.

In Figure 1 f. & g., the results are consistent with a shift in the work function and not changes due to coordination. All of the peaks move together and by the same amount. I think that the interpretation of coordination between the sulfur and under coordinated Pb is erroneous. If coordination was being observed, then the shifts would have gone in opposite directions (Pb to lower energy and S to higher energy).

Page 6, line 107: The authors have written "The reduction in binding energy observed for the sulfur atom within the C-S bond in the PCBM@TMDS sample suggests a decrease in the electron cloud density surrounding sulfur, thus confirming the electron transfer between sulfur radicals and PCBM". What I can see in Fig 1e is that the binding energy of -C-S in TMDS is 164.50 and in PCBM@TMDS is 164.65 eV which shows as increase. Please clarify it.

Page 6, line 113: The authors have mentioned that they solely coated the TMDS material on the surface of the perovskite and characterized the samples using XPS. Regarding that I wonder if only the coordinated TMDS can convert to radicals and the corresponding electrons lead n-doping? Since mass spectrum also show unreacted TMDS, I think the authors need to explain more about the mechanism. Do they consider any stoichiometric reactions? Do they investigated if the amount of TMDS can affect on the final performance? Will the radicals stay connected to the Pb atoms after irradiation? How will the new PCBM connect to the surface? Why they have chosen the mixed CsPb perovskite?

Supplementary Table 2 and 3: the authors should write which equation they have use to calculate tau. The values seems higher compared to tau1 and the amplitudes.

It is not clear how the authors calculated the average time constants for the time-resolved photoluminescence. By considering the standard weighted average equation ($\tau_{avg} = A_1\tau_1 + A_2\tau_2$) the results of table 2 will be 286.5 and 482.3 and for table 3 will be 207.39 and 233.3 respectively, which shows the different trend!

Page 13 and Fig 4a: I wonder why authors talked about reverse scan parameters but when it comes to the certified PCE they talked about forward scan results.

Moreover there are some inconsistency between the pictures, tables and the text that I ask the authors to check all values once more. For example:

Fig 3c: While the trap densities were written as $8.84 \times 10^{14} \text{ cm}^{-3}$ and $1.55 \times 10^{15} \text{ cm}^{-3}$ in the text (page 11), but they are written as 8.84×10^{-14} and 55×10^{-15} in Fig 3.

Supplementary Table 1: ECB for target should be minus (-4.06).

Reviewer #4

(Remarks to the Author)

Version 1:

Reviewer comments:

Reviewer #1

(Remarks to the Author)

The revised manuscript has addressed my concerns. Therefore, I recommend publication essentially as it stands.

Reviewer #2

(Remarks to the Author)

It is acceptable.

Reviewer #3

(Remarks to the Author)

The authors have sufficiently addressed my concerns and the work can be published.

Reviewer #4

(Remarks to the Author)

Response Letter

We acknowledge referees' insightful and professional comments and suggestions very much, which are valuable in improving the quality of our manuscript. Here we have addressed the queries from the reviewers point by point and revised the manuscript according to the comments. The revision was highlighted in red font in the revised manuscript.

Reviewer #1 (Remarks to the Author)

Comments: In this manuscript, the authors reported a strategy to enhance the electrical properties and surface morphology of PCBM films by introducing TMDS into PCBM solution to generate reducing radicals under UV irradiation condition. The electron acceptor PCBM absorbed the free electrons from these sulfur radicals, leading to the n doping effect in PCBM. As a result, this approach yielded a device efficiency of 26.10% (certified 25.56%) without hysteresis for the target device. Moreover, a 1 cm² device achieved an efficiency of 24.06%. More importantly, the target device maintained over 95% and 90.4% of its initial efficiency even after 1,271 h of MPPT and 1,090 h at a temperature of 85 °C and RH of 85%, respectively. This manuscript can be accepted for publication after addressing the following issues.

Reply: We thank the reviewer for positive evaluation and recognition on our work very much. We also appreciate the reviewer's valuable and constructive comments to help us improve the manuscript quality.

1. It would be more appropriate to change the title to "Highly efficient and stable inverted perovskite solar cells enabled by homogenized PCBM with enhanced electron transport" to specify the used fullerene derivative.

Reply: We gratefully appreciate the valuable comments. The title is changed to "Highly efficient and stable inverted perovskite solar cells enabled by homogenized PCBM with enhanced electron transport".

2. The use of the "@" in "PCBM@TMDS" is incorrect and should be modified.

Reply: We thank the reviewer for the professional comments. “PCBM@TMDS” is corrected to “PCBM with TMDS”.

3. In the device fabrication, a solution of BCP in IPA with a concentration of 5 mg/mL was used, which should be checked, because the concentration in most cases in the literature was 0.5 mg/mL.

Reply: We agree with the reviewer’s comments that BCP is not soluble at 5 mg/ml concentration in IPA. Here 5 mg BCP was added into 1 ml IPA to prepare a supersaturated solution, which was filtered by PTFE filter before use (Fig. R1). The relevant details of BCP deposition have been revised in Method.

Fig.R1 The supersaturated solution (5 mg/ml) of BCP (left) and the solution filtered using a PTFE filter (right).

Revisions in the revised manuscript:

Line 368-369, Page 19:

Afterwards, 5 mg BCP was added into 1 mL IPA to prepare a supersaturated solution, which was filtered by PTFE filter before use. Afterward, the obtained saturated solution was spin-coated on PCBM film at 5000 rpm for 30 s.

4. The related references of “Energy Environ. Sci., 2016, 9, 3424; Adv. Energy Mater. 2023, 13, 2302191; Nat. Commun. 2023, 14, 8052; Adv. Mater. 2024, 36, 2308706.” can be cited.

Reply: We thank the reviewer for the constructive comments. These references are appropriately cited in the revised manuscript.

Revisions in the revised manuscript:

References

19. Wang WF, et al. Electrosynthesis of buckyballs with fused-ring systems from PCBM and its analogue. *Nat. Commun.* 14, 8052 (2023).
20. Sun HF, et al. Scalable solution-processed hybrid electron transport layers for efficient all-perovskite tandem solar modules. *Adv. Mater.* 36, 2308706 (2024).
22. Bin ZY, Li JW, Wang LD, Duan L. Efficient n-type dopants with extremely low doping ratios for high performance inverted perovskite solar cells. *Energy Environ. Sci.* 9, 3424-3428 (2016).
31. Sun XH, et al. V_{oc} of inverted perovskite solar cells based on N-doped PCBM exceeds 1.2 V: interface energy alignment and synergistic passivation. *Adv. Energy Mater.* 13, 2302191 (2023).

Reviewer #2 (Remarks to the Author):

Comments: The authors conducted investigations to reveal the importance of the method to improve the properties of PCBM using TMDS in inverted PSCs. Through TMDS treatment, the formation of reducing sulfur radicals could be introduced into the PCBM to achieve n doping, which improves its electrical conductivity and electron mobility. The results show that the TMDS-treated PCBM layer exhibits excellent properties, resulting in the TMDS device with superior photovoltaic performance. This work is of interest to the research community of inverted PSCs, which may be considered for publication if the following issues can be addressed:

Reply: We thank the reviewer for positive evaluation and recognition on our work. We also greatly appreciate the reviewer's valuable and constructive comments to help us improve the manuscript quality.

Comments of improvement:

1. In the introductory section, the authors claim that "C₆₀ is insoluble in many solvents and requires expensive thermal evaporation procedures for processing, restricting its application in all-solution-processed PSCs. This issue has led to the widespread utilization of PCBM". I think this statement is wrong, the material cost of C₆₀ is much lower than PCBM in the current trend of large-area module preparation. The C₆₀ vapour deposition method has already reached the industrial consensus, which can help the PSCs devices to achieve large-area module preparation. Therefore, the authors need to further elaborate on the significance of modifying PCBM rather than C₆₀.

Reply: We thank the reviewer for the valuable comments very much. We have made the corresponding modifications to the description in the introductory section as shown below.

Revisions in the revised manuscript:

(1) Line 43-47, Page 3:

In inverted PSCs, fullerene derivatives such as [6,6]-phenyl-C₆₁-butyric acid methyl ester (PCBM) or C₆₀ are commonly employed as electron transport layers (ETLs)^{9,11,12}. Compared to C₆₀, PCBM possesses a

phenylbutanoate methyl ester group, which effectively reduces defects on the perovskite surface and minimizes carrier loss⁹. Simultaneously, PCBM exhibits better interface compatibility with the perovskite, which is beneficial for fabricating high-performance devices. The remarkable electron transport capability of PCBM allows for efficient electron extraction from the perovskite layer. Nonetheless, its intrinsic stability remains a critical factor influencing device performance and stability.

2. The methodology for specific sampling process and electron spin resonance (ESR) measurement are not described.

Reply: Thanks for your valuable comments. The methodology for the specific sampling process and electron spin resonance (ESR) measurement is added in the Methods section.

Revisions in the revised manuscript:

(1) Line 428-434, Page 21:

Short-circuit conditions were established by directly connecting the Ag electrode and the ITO electrode. The ESR measurements had been performed on a Bruker EMXnano band pulsed-ESR spectrometer. A standard sample of the magnetic field (sodium thiosulfate pentahydrate) was used to calibrate the magnetic field. For the PCBM solution, 23 mg of PCBM had been added to 1 mL of CB solution; for the PCBM modified by TMDS solution, 23 mg of PCBM and 0.48 mg (3×10^{-3} mmol/mL) of TMDS had been added to 1 mL of CB solution. Samples requiring UV light treatment were irradiated with 365 nm ultraviolet light for 2 hours.

3. The authors demonstrated the interaction between TMDS and PCBM by XPS, and I am curious whether TMDS would diffuse into the phase or the grain boundaries, which is also important for PSCs. So, I would suggest the authors try TOF-SIMS, which may help better characterize the spatial distribution of TMDS.

Reply: We thank the reviewer for the professional comments. The ToF-SIMS spectrum of the device cross-section has been added (Supplementary Fig. 5). It can be observed that TMDS diffuses into the perovskite layers, exhibiting a gradient distribution. The relevant discussion has been included in the revised manuscript.

Supplementary Fig. 5. ToF-SIMS depth results for TMDS modified devices.

Revisions in the revised manuscript:

(1) Line 166-168 Page 9:

The target PCBM forms more conformal contacts with the perovskite due to the coordinating effect between its TMDS and uncoordinated Pb²⁺ on the perovskite surface, resulting in a more uniform morphology of the PCBM film⁹. Additionally, the ToF-SIMS spectrum also shows that TMDS can diffuse into the perovskite layer, displaying a gradient distribution (Supplementary Fig. 5). This facilitates the deep passivation of TMDS within the perovskite layer.

4. The authors mention on page 8 that "the homogeneous particle size is attributed to the strong interaction between TMDS and PCBM, which prevents the aggregation of more than a single unit of particles in solution." What type of interaction between TMDS and PCBM occurs? There seems to be a lack of corresponding evidence. Please provide some additional evidence such as FTIR or DFT calculations.

Reply: We thank the reviewer for the professional comment. After UV irradiation, free radicals generated by TMDS induce n doping in PCBM, as evidenced by Fig. 1d. After the charge transfer process, the organic radicals may become positively charged dipoles by losing electrons, while PCBM becomes negatively charged dipoles by gaining electrons. Consequently, this facilitates the formation of dipole-dipole interactions between them, which would contribute to stabilizing the structure of the complex (*Energy Environ. Sci.*, 15, 2096-2107, 2022). To explore this further, we conduct FTIR and UV absorption measurements. After UV light irradiation, characteristic peaks of the S-S and C=S functional groups are still observed in the PCBM with TMDS sample (Supplementary Fig. 1), indicating that unreacted TMDS monomers remain after UV light irradiation. Additionally, a new peak appears at 839.8 cm^{-1} in the FTIR spectrum of the PCBM with TMDS sample (Supplementary Fig. 2a). Similarly, in the UV-vis absorption spectra, the PCBM with TMDS sample exhibits a new peak at 480 nm (Supplementary Fig. 2b). The relevant discussion has been incorporated into the manuscript.

Supplementary Fig. 1. a, b Fourier transforms infrared (FTIR) spectra of TMDS and PCBM films without and with TMDS.

Supplementary Fig. 2. a, FTIR spectra of TMDS and PCBM films without and with TMDS. **b**, UV-vis absorption spectra of the PCBM and PCBM with TMDS film.

Revisions in the revised manuscript:

Line 111-120, Page 6:

The reduction in binding energy observed for the sulfur atom within the C-S bond in the PCBM with TMDS sample suggests a decrease in the electron cloud density surrounding sulfur, thus confirming the electron transfer between sulfur radicals and PCBM. After UV light irradiation, characteristic peaks of the S-S and C=S functional groups are still observed in the PCBM with TMDS sample (Supplementary Fig. 1), indicating that unreacted TMDS monomers remain after UV light irradiation. Additionally, a new peak appears at 839.8 cm⁻¹ in the FTIR spectrum of the PCBM with TMDS sample (Supplementary Fig. 2a). Similarly, in the UV-vis absorption spectra, the PCBM with TMDS sample exhibits a new peak at 480 nm (Supplementary Fig. 2b). Combining the results of FTIR and UV-vis absorption spectra, it is suggested that after the charge transfer process, the organic radicals may become positively charged dipoles upon losing electrons, while PCBM becomes negatively charged dipoles upon gaining electrons. Therefore, this promotes the formation of dipole-dipole interactions between them, which helps to stabilize the structure of the complex²⁶.

References

26. Hu SF, et al. Optimized carrier extraction at interfaces for 23.6% efficient tin-lead perovskite solar cells. *Energy Environ. Sci.* **15**, 2096-2107 (2022).

5. The PSCs fabricated by the spin coating method have achieved an efficiency of over 24% on devices with an area of 1 cm². However, the spin coating process is inherently not scalable. If possible, the authors should demonstrate the compatibility of this method with scalable fabrication techniques, such as blade coating, slot-die coating, spray coating, etc. We are also curious about the performance of TMDS-PCBM in large-area module devices.

Reply: We thank the reviewer for the professional comment. We evaluate the compatibility of the TMDS additive with upscaling processes by fabricating perovskite modules on a 30 cm × 30 cm substrate. As shown in Supplementary Fig. 16, the active area of module is 669 cm² in this context. The PCE of the champion module is 17.05% and 15.30% for target and control devices, respectively. The preparation method of the module has been added to the revised manuscript.

Supplementary Fig. 16. J–V curves of the module with relevant photovoltaic parameters and photographic image depicted in the insets.

Revisions in the revised manuscript:

Line 373-385, Page 19:

For the modules, first, use Physical Vapor Deposition (PVD) technology to prepare NiO_x with a thickness of approximately 18 nm on the FTO substrate. The FA_{0.83}Cs_{0.17}PbI₃ perovskite film is prepared by dissolving 0.2086 g of PbCl₂, 1.325 g of CsI, 0.3038 g of MAI, 15.213 g of PbI₂, and 4.282 g of FAI in a 30 ml mixed solvent solution (v/v, DMF : NMP : ACN = 6 : 1 : 3) with a concentration of 1.1 mmol/mL. The perovskite precursor solution is then slot-die coated onto a glass/FTO/NiO_x substrate with a slot-die gap of 100 μm above the substrate at a speed of 5 mm/s. Then, the resulting wet perovskite films are quickly transferred into a vacuum chamber, which is pumped to 10 Pa and maintained for 40 s. The resulting perovskite films are then annealed at 150 °C for 15 minutes in the air (RH=20%). Subsequently, the PCBM solution (23 mg/mL) is then slot-die coated onto the perovskite film with a slot-die gap of 100 μm at a speed of 4 mm/s. 20 nm of ALD-SnO₂ is deposited on the PCBM surface. Finally, the ITO/Cu/ITO sandwich electrodes are prepared on the ALD-SnO₂ surface using PVD technology, with thicknesses of 20 nm/15 nm/20 nm, respectively.

6. It is strongly recommended the authors use a more complete physical model for the TRPL to demonstrate optimized extraction of charges (e.g. computed differential lifetimes from fits to the transients). The full decay is governed by nonradiative, trap-assisted surface/bulk recombination (mostly monoexponential decay), radiative recombination (“bimolecular,” second-order decay), and charge transfer effects. In this manuscript, the introduction of TMDS also may cause the passivation of defects, which also change decay lifetimes. Consequently, the above analysis of the PL mapping results and the TRPL lifetime results appears to be inadequate. It is recommended that the authors incorporate light intensity-dependent PLQY or hyperspectral techniques to account for the above. It is strongly recommended the authors cite some important relevant literature (e.g. *Science* 370,1300-1309 (2020), *Matter* 7, 1054–1070 (2024)).

Reply: We thank the reviewer for the valuable suggestions. As suggested by referee, we have provided the calculated differential lifetimes from fits to the transients in Fig. 3i based on the model proposed by Krogmeier et al (*Sustainable Energy Fuels*, 2, 1027-1034, 2018; *Science*, 370,1300-1309, 2020; *Matter*, 7, 1054–1070, 2024)

(Supplementary Fig. 9). We also added the corresponding discussion in the main text. The TRPL measurements have been conducted at multiple excitation intensities. As shown in Supplementary Fig. 10, the carrier lifetime was increased after incorporation of TMDS and the same tendency was observed at different excitation intensities. However, it is worth to note that the charge carrier lifetime was also reduced slightly with the increase of excitation intensity due to the increase of bimolecular recombination losses (*Nature*, 555, 497-501, 2018). Meanwhile, the variation of carrier lifetime of target samples depending on excitation intensity was not as significant as that of control sample, which is indicative of the lower defect density in the target samples. In addition, we have cited both of these important relevant literatures in the revised manuscript.

Supplementary Fig. 9. Calculated differential lifetimes from fits to the transients in Fig. 3i.

Supplementary Fig. 10. a, b, Intensity dependent time-resolved photoluminescence decays of the PVSK/PCBM (a) and PVSK/PCBM with TMDS (b).

Revisions in the revised manuscript:

Line 229-241, Page 12:

The decrease in the PL intensity and shortened carrier lifetime also confirm the optimized extraction of charges (Figs. 3h, 3i, and Supplementary Table 3). Fits to the TRPL transients were used to compute the differential lifetime using Krogmeier et al.'s model³⁴⁻³⁶ (Supplementary Fig. 9). The charge transfer process at early times (~250 ns) led to a faster rise of differential lifetime in target film than that of control film. The transition from increasing lifetime to the plateau marks the end of charge transfer, and non-radiative first-order recombination becomes dominant. The TRPL measurements have been conducted at multiple excitation intensities. As shown in Supplementary Fig. 10, the carrier lifetime was increased after incorporation of TMDS and the same tendency was observed at different excitation intensities. However, it is worth to note that the charge carrier lifetime was also reduced slightly with the increase of excitation intensity due to the increase of bimolecular recombination losses³⁷. Meanwhile, the variation of carrier lifetime of target samples depending on excitation intensity was not as significant as that of control sample, which is indicative of the lower defect density in the target samples.

References

34. Luo XY, et al. Effects of local compositional heterogeneity in mixed halide perovskites on blue electroluminescence. *Matter* **7**, 1054-1070 (2024).
35. Al-Ashouri A, et al. Monolithic perovskite/silicon tandem solar cell with >29% efficiency by enhanced hole extraction. *Science* **370**, 1300-1309 (2020).
36. Krogmeier B, Staub F, Grabowski D, Rau U, Kirchartz T. Quantitative analysis of the transient photoluminescence of CH₃NH₃PbI₃/PC₆₁BM heterojunctions by numerical simulations. *Sustainable Energy Fuels* **2**, 1027-1034 (2018).

7. Some typographic errors should be corrected, such as Supplementary Fig. 7 and caption, lines 151.

Reply: We thank the reviewer for the valuable suggestions. The typographic errors in Supplementary Fig. 8 and caption in line 170 have been corrected. We also checked the manuscript and supporting information again to avoid similar errors.

Revisions in the revised Supplementary Information:

Line 170, Page 9:

Supplementary Fig. 8. a, X-ray diffraction (XRD) patterns of the pristine and TMDS modified perovskite films.

b, c, XRD patterns and full width at half maximum (FWHM) for control (**b**) and target (**c**) perovskite films.

Reviewer #3 (Remarks to the Author)

Comments: The communication entitled “Highly efficient and stable inverted perovskite solar cells enabled by homogenized fullerene with enhanced electron transport” written by Cheng Gong, et. al. has focused on introducing tetramethylthiuram disulfide (TMDS) into PCBM. Under UV light highly reducing organic materials produced in solution which helped to suppress charge recombination and increase conductivity and electron mobility, and performance of inverted perovskite solar cells compare to classical electron acceptor PCBM. While the work interesting, some of the core claims of the article are lacking sufficient justification to recommend publication.

Reply: We appreciate the reviewer’s valuable comments and suggestions very much. We have carefully revised our manuscript according to your proposed suggestions. We sincerely hope that the revised manuscript will meet the publication requirement.

1. While the TMDS generates radicals and transfers at least some of them to PCBM, it is not clear that this “doping” strategy results in long lasting effects for the PCBM. If the result is actually doping, then the radicals that are generated should be detectable on a long-term time scale. Are PCBM radicals detectable by EPR after 1000 hr, which is a similar time for the solar cell stability tests?

Reply: We gratefully appreciate the valuable comments. The solution of TMDS-doped PCBM still exhibits a free radical signal after 1000 hours (Fig. 1d), which is similar to the stability duration of the devices. The relevant result has been added into the manuscript.

Fig.1 d, The ESR spectra of different solutions.

2. Figure 1d: Please provide ESR of the solution containing only the TMDS after UV radiation. Then it could be easier to conclude if the paramagnetic signal is from fullerene radical or any other radical due to radical reactions in TMDS.

Reply: We gratefully appreciate the valuable comments. The TMDS sample does not exhibit an obvious free radical signal after UV light irradiation (Fig. 1d). This suggests that the signal produced by the PCBM with TMDS sample following UV light irradiation is attributed to the fullerene radical.

Fig.1 d, The ESR spectra of different solutions.

3. Figure 1e: the authors claimed that S-S bond dissociation happens under UV irradiation. I wonder why we can still see S-S bond curve even with more intensity after irradiation? While Fig S1 also show only a weak peak in FTIR compare to the TMDS. Further, there is not obviously a significant difference between the data for TMDS+UV and TMDS+PCBM+UV. The data are not sufficiently clean to make the claims that the authors are making.

Reply: We gratefully appreciate the professional comments. After UV irradiation, free radicals generated by TMDS induce n doping in PCBM, as evidenced by Fig. 1d. After the charge transfer process, the organic radicals may become positively charged dipoles by losing electrons, while PCBM becomes negatively charged dipoles by gaining electrons. Consequently, this facilitates the formation of dipole-dipole interactions between them, which would contribute to stabilizing the structure of the complex (*Energy Environ. Sci.* 15, 2096-2107, 2022). To explore this further, we conduct FTIR and UV absorption measurements. After UV light irradiation, characteristic peaks of the S-S and C=S functional groups are still observed in the PCBM with TMDS sample (Supplementary Fig. 1), indicating that unreacted TMDS monomers remain after UV light irradiation. Additionally, a new peak appears at 839.8 cm^{-1} in the FTIR spectrum of the PCBM with TMDS sample (Supplementary Fig. 2a). Similarly, in the UV-vis absorption spectra (Supplementary Fig. 2b), the PCBM with TMDS sample exhibits a new peak at 480 nm. All of these indicate the formation of n doping effects in PCBM. The relevant discussion has been incorporated into the manuscript.

Supplementary Fig. 1. a, b Fourier transforms infrared (FTIR) spectra of TMDS and PCBM films without and with TMDS.

Supplementary Fig. 2. a, b FTIR spectra of TMDS and PCBM films without and with TMDS. **b,** UV-vis absorption spectra of the PCBM and PCBM with TMDS film.

Revisions in the revised manuscript:

Line 111-120, Page 6:

The reduction in binding energy observed for the sulfur atom within the C-S bond in the PCBM with TMDS sample suggests a decrease in the electron cloud density surrounding sulfur, thus confirming the electron transfer between sulfur radicals and PCBM. After UV light irradiation, characteristic peaks of the S-S and C=S functional groups are still observed in the PCBM with TMDS sample (Supplementary Fig. 1), indicating that unreacted TMDS monomers remain after UV light irradiation. Additionally, a new peak appears at 839.8 cm⁻¹ in the FTIR spectrum of the PCBM with TMDS sample (Supplementary Fig. 2a). Similarly, in the UV-vis absorption spectra, the PCBM with TMDS sample exhibits a new peak at 480 nm (Supplementary Fig. 2b). Combining the results of FTIR and UV-vis absorption spectra, it is suggested that after the charge transfer process, the organic radicals may become positively charged dipoles upon losing electrons, while PCBM becomes negatively charged dipoles upon gaining electrons. Therefore, this promotes the formation of dipole-dipole interactions between them, which helps to stabilize the structure of the complex²⁶.

4. In Figure 1 f. & g., the results are consistent with a shift in the work function and not changes due to coordination. All of the peaks move together and by the same amount. I think that the interpretation of coordination between the sulfur and under coordinated Pb is erroneous. If coordination was being observed, then the shifts would have gone in opposite directions (Pb to lower energy and S to higher energy).

Reply: We gratefully appreciate the professional comments. Fig. 1g contains an error due to a lack of calibration.

We have recalibrated the XPS spectrum as shown in Fig. 1g.

Fig. 1. g, XPS spectra of Pb 4f for TMDS and PVSK films without and with TMDS

Revisions in the revised manuscript:

Line 127, Page 6:

Correspondingly, the characteristic peak of Pb 4f also shifts toward a **lower** binding energy by **0.45 eV**, suggesting effective coordination between uncoordinated Pb²⁺ on the perovskite surface and -C=S in TMDS (Fig. 1g).

5. Page 6, line 107: The authors have written ‘‘The reduction in binding energy observed for the sulfur atom within the C-S bond in the PCBM@TMDS sample suggests a decrease in the electron cloud density surrounding sulfur, thus confirming the electron transfer between sulfur radicals and PCBM’’. What I can see in Fig 1e is that

the binding energy of -C-S in TMDS is 164.50 and in PCBM@TMDS is 164.65 eV which shows as increase.

Please clarify it.

Reply: Thank you for your professional comments. This is a textual error; “reduction” should be replaced with “increase.” In XPS measurement, an increase in binding energy typically indicates a decrease in charge density (*Science*, 382, 1399-1404, 2023). We apologize for the oversight and have made the correction in the revised manuscript.

Revisions in the revised manuscript:

Line 108, Page 6:

The **increase** in binding energy observed for the sulfur atom within the C-S bond in the PCBM with TMDS sample suggests a decrease in the electron cloud density surrounding sulfur, thus confirming the electron transfer between sulfur radicals and PCBM.

5. Page 6, line 113: The authors have mentioned that they solely coated the TMDS material on the surface of the perovskite and characterized the samples using XPS. Regarding that I wonder if only the coordinated TMDS can convert to radicals and the corresponding electrons lead n-doping? Since mass spectrum also show unreacted TMDS, I think the authors need to explain more about the mechanism. Do they consider any stoichiometric reactions? Do they investigated if the amount of TMDS can affect on the final performance? Will the radicals stay connected to the Pb atoms after irradiation? How will the new PCBM connect to the surface? Why they have chosen the mixed CsPb perovskite?

Reply: Thank you for your professional comments. In this work, TMDS is mixed with PCBM and then subjected to UV light treatment. Therefore, the actual situation is that TMDS molecules are first converted into radicals, thereby forming an n doping effect on PCBM. After UV light exposure, TMDS cannot be completely converted into radicals, and there is still a portion of unreacted TMDS molecules present in the PCBM solution. As a result, signals of unreacted TMDS can still be observed in the mass spectrometry results. Generally speaking, the peak

intensity ratio in mass spectrometry can be used to calculate the relative content of components and therefore determine the conversion rate of a reaction. We performed mass spectrometry analysis on the films prepared from PCBM modified by TMDS solutions with different illumination times. I_{Radicals} represents the intensity of free radicals, while I_{TMDS} represents the intensity of unreacted TMDS (Table R1). It can be seen that within 2 hours of UV illumination, the conversion rate of free radicals gradually increases, but after 2 hours, the ratio of $I_{\text{Radicals}}/I_{\text{TMDS}}$ remains constant at 3.1 regardless of reaction time. This indicates that after 2 hours of illumination, the conversion rate for TMDS to produce free radicals is fixed. When the content of TMDS is fixed, the amount of free radicals it generates is also fixed.

The conductivity of PCBM modified with different concentrations of TMDS is shown in Supplementary Fig. 5. The conductivity reaches its maximum when the TMDS concentration is 2×10^{-3} mmol/ml. We supplement this with statistical data on the device efficiency modified with varying TMDS concentrations. It can be seen that the average efficiency of the device is highest when the TMDS concentration is 2×10^{-3} mmol/ml. In Figs. 1f and g, it is observed that the functional group capable of coordinating with Pb is -C=S, which is present in both the radicals and the TMDS molecules. Therefore, despite UV irradiation, the radicals can form coordination bonds with Pb. In Comment 1, the dipole interaction between radicals and PCBM has already been confirmed. Since the radicals can also form coordination bonds with Pb, their presence enhances the connection between PCBM and the perovskite layer, facilitating a more uniform coverage of the PCBM film. α -FAPbI₃ perovskite, due to its narrow bandgap (1.48 eV), has a broader spectral response range and significant photovoltaic potential. However, α -FAPbI₃ perovskite is prone to phase transition at room temperature, converting to the non-optically active δ -phase. Therefore, in this work, we use a small amount of Cs⁺ to partially replace FA⁺ to stabilize the black phase of α -FAPbI₃ perovskite. We choose the CsFA system because it maintains a narrow bandgap while being stable at room temperature. This is highly beneficial for the fabrication of high-performance devices. As a result, the device's short-circuit current density exceeds 26 mA/cm².

Table R1. The mass spectrum peak intensity ratio of PCBM with TMDS films prepared under different UV exposure times.

	10 min	30 min	1 hour	2 hours	4 hours	8 hours
$I_{\text{Radicals}}/I_{\text{TMDS}}$	1.8	2.1	2.6	3.1	3.1	3.1

Fig R2. Statistical distribution diagram of the photovoltaic parameters of the control and various concentrations TMDS modified devices. The statistical data were obtained from 15 individual cells for each kind of device.

6. Supplementary Table 2 and 3: the authors should write which equation they have use dto calculate tave. The values seems higher compared to t1 an t1 and the amplitudes. It is not clear how the authors calculated the average time constants for the time-resolved photoluminescence. By considering the standard weighted average equation ($t_{avg}=A_1t_1+A_2t_2$) the results of table 2 will be 286.5 and 482.3 and for table 3 will be 207.39 and 233.3 respectively, which shows the different trend!

Reply: Thank you for your professional comments. The average weighted lifetime is extracted using the equation $\tau_{ave} = (A_1\tau_1^2 + A_2\tau_2^2)/(A_1\tau_1 + A_2\tau_2)$. In Supplementary Table 3, there are textual errors in the values of A_1

and A_2 , which have now been corrected as shown. The opposite trends in Supplementary Table 2 and Supplementary Table 3 occur because the former sample is prepared on a pure glass substrate without a transport layer, while the latter sample includes an electron transport layer. The increase in lifetime for the sample on the pure glass substrate after TMDS modification is due to the defect passivation effect of TMDS. In contrast, with the presence of a transport layer, the lifetime of the target sample is shorter because the TMDS-modified PCBM facilitates faster electron extraction and transfer. In this manuscript, the introduction of TMDS also may cause the passivation of defects, which also change decay lifetimes. Consequently, the above analysis of the PL results and the TRPL lifetime results appears to be inadequate.

Therefore, we use a more complete physical model for the TRPL to demonstrate optimized extraction of charges (e.g. calculated differential lifetimes from fits to the transients). We have provided the calculated differential lifetimes from fits to the transients in Fig. 3i based on the model proposed by Krogmeier et al (*Sustainable Energy Fuels*, 2, 1027-1034, 2018) (Supplementary Fig. 9). We also added the corresponding discussion in the main text. The TRPL measurements have been conducted at multiple excitation intensities. As shown in Fig. 10, the carrier lifetime was increased after incorporation of TMDS and the same tendency was observed at different excitation intensities. However, it is worth to note that the charge carrier lifetime was also reduced slightly with the increase of excitation intensity due to the increase of bimolecular recombination losses (*Nature*, 555, 497-501, 2018). Meanwhile, the variation of carrier lifetime of target samples depending on excitation intensity was not as significant as that of control sample, which is indicative of the lower defect density in the target samples.

Supplementary Table 3. Fitting results of TRPL curves of the glass/PVSK/PCBM without and with TMDS.

Sample	A_1	τ_1 (ns)	A_2	τ_2 (ns)	τ_{avg} (ns)
PVSK/PCBM	0.7	102.1	0.3	453.3	332.3
PVSK/PCBM with TMDS	0.93	37.5	0.07	527.0	289.1

Note: The average weighted lifetime is extracted using the equation $\tau_{\text{ave}} = (A_1\tau_1^2 + A_2\tau_2^2)/(A_1\tau_1 + A_2\tau_2)$

Supplementary Fig. 9. Calculated differential lifetimes from fits to the transients in Fig. 3i.

Supplementary Fig. 10. a, b, Intensity dependent time-resolved photoluminescence decays of the PVSK/PCBM (a) and PVSK/PCBM with TMDS (b).

Revisions in the revised manuscript:

Line 228-233, Page 12:

The decrease in the PL intensity and shortened carrier lifetime also confirm the optimized extraction of charges (Figs. 3h, 3i, and Supplementary Table 3). Fits to the TRPL transients were used to compute the differential lifetime using Krogmeier et al.'s model³⁴⁻³⁶ (Supplementary Fig. 9). The charge transfer process at early times

(~250 ns) led to a faster rise of differential lifetime in target film than that of control film. The transition from increasing lifetime to the plateau marks the end of charge transfer, and non-radiative first-order recombination becomes dominant. The TRPL measurements have been conducted at multiple excitation intensities. As shown in Supplementary Fig. 10, the carrier lifetime was increased after incorporation of TMDS and the same tendency was observed at different excitation intensities. However, it is worth to note that the charge carrier lifetime was also reduced slightly with the increase of excitation intensity due to the increase of bimolecular recombination losses³⁷. Meanwhile, the variation of carrier lifetime of target samples depending on excitation intensity was not as significant as that of control sample, which is indicative of the lower defect density in the target samples.

References

34. Luo XY, et al. Effects of local compositional heterogeneity in mixed halide perovskites on blue electroluminescence. *Matter* **7**, 1054-1070 (2024).
35. Al-Ashouri A, et al. Monolithic perovskite/silicon tandem solar cell with >29% efficiency by enhanced hole extraction. *Science* **370**, 1300-1309 (2020).
36. Krogmeier B, Staub F, Grabowski D, Rau U, Kirchartz T. Quantitative analysis of the transient photoluminescence of CH₃NH₃PbI₃/PC₆₁BM heterojunctions by numerical simulations. *Sustainable Energy Fuels* **2**, 1027-1034 (2018).

7. Page 13 and Fig 4a: I wonder why authors talked about reverse scan parameters but when it comes to the certified PCE they talked about forward scan results. Moreover, there are some inconsistency between the pictures, tables and the text that I ask the authors to check all values once more. For example: Fig 3c: While the trap densities were written as $8.84 \times 10^{14} \text{ cm}^{-3}$ and $1.55 \times 10^{15} \text{ cm}^{-3}$ in the text (page 11), but they are written as 8.84×10^{14} and 55×10^{15} in Fig 3. Supplementary Table 1: ECB for target should be minus (-4.06).

Reply: Thank you for your professional comments. We previously focused only on the efficiency values of the devices and overlooked the discussion of the consistency between certified PCE and laboratory-measured

efficiency. To ensure uniformity, we discuss the reverse scan parameters of the certified PCE in the revised manuscript, which aligns with the statements in other J - V curves.

The trap densities in Fig. 3c (which should be $8.84 \times 10^{14} \text{ cm}^{-3}$ and $1.55 \times 10^{15} \text{ cm}^{-3}$) and the E_{CB} in Supplementary Table 1 have been corrected in. Additionally, we have conducted a thorough review of the revised manuscript to avoid similar errors.

Revisions in the revised manuscript:

Fig. 3c SCLC plots of the electron-only device ITO/SnO₂/perovskite/PCBM/Ag where the PCBM films without and with TMDS were used.

Revisions in the revised manuscript:

Line 359, Page 20:

Supplementary Table 1. Calculated valence band (E_{VB}) and conduction band (E_{CB}) from $E_{cut-off}$, E_F and E_g for the PCBM films without and with TMDS.

Sample	$E_{cut-off}$ (eV)	E_{on-set} (eV)	E_{F-edge} (eV)	E_{VB} (eV)	E_g (eV)	E_{CB} (eV)
Target	16.17	1.17	-5.05	-6.22	2.16	-4.06
Control	15.79	1.04	-5.43	-6.47	2.16	-4.31

Reviewer #4 (Remarks to the Author)

Comments: I co-reviewed this manuscript with one of the reviewers who provided the listed reports. This is part of the Nature Communications initiative to facilitate training in peer review and to provide appropriate recognition for Early Career Researchers who co-review manuscripts.

Reply: We appreciate the reviewer's valuable comments and suggestions. We have carefully revised our manuscript according to the proposed suggestions. We sincerely hope that the revised manuscript will meet the publication requirement.